# Characteristics and Roles of T Follicular Helper Cells in SARS-CoV-2 Vaccine Response

**DOI:** 10.3390/vaccines10101623

**Published:** 2022-09-28

**Authors:** Xuyang Chi, Jia Gu, Xiaoxue Ma

**Affiliations:** 1Department of Pediatrics, The First Hospital of China Medical University, Shenyang 110001, China; 2Department of Microbiology & Immunology and Pediatrics, Dalhousie University, and Canadian Center for Vaccinology, IWK Health Centre, Halifax, NS B3K 6R8, Canada

**Keywords:** SARS-CoV-2, vaccine, Tfh cells, immunoglobulin

## Abstract

Severe acute respiratory syndrome coronavirus 2 (SARS-CoV-2) vaccination is critical to controlling the coronavirus disease 2019 (COVID-19) pandemic. However, a weak response to the vaccine and insufficient persistence of specific antibodies may threaten the global impact of mass vaccination campaigns. This study summarizes the internal factors of the body that affect the effectiveness of the SARS-CoV-2 vaccine. T follicular helper (Tfh) cells support germinal center B cells to produce vaccine-specific immunoglobulins. A reduction in the Tfh cell number and a shift in the subset phenotypes caused by multiple factors may impair the production and persistence of high-affinity antibodies. Besides efficacy differences caused by the different types of vaccines, the factors that affect vaccine effectiveness by intervening in the Tfh cell response also include age-related defects, the polarity of the body microenvironment, repeated immunization, immunodeficiency, and immunosuppressive treatments. Assessing the phenotypic distribution and activation levels of Tfh cell subsets after vaccination is helpful in predicting vaccine responses and may identify potential targets for improving vaccine effectiveness.

## 1. Introduction

The emergence of severe acute respiratory syndrome coronavirus 2 (SARS-CoV-2) has caused a large global disease outbreak and the coronavirus disease 2019 (COVID-19) pandemic. There have been over 540 million confirmed cases and close to 6.3 million deaths worldwide (https://covid19.who.int, accessed on 1 August 2022). The pandemic has caused serious financial, socio-economical, and political problems. Although specific drugs to treat SARS-CoV-2 have been identified, vaccination is still urgently needed to limit the spread of the disease and prevent severe cases.

Vaccination is now an indispensable medical intervention that has saved millions of lives. Most SARS-CoV-2 vaccines have been studied in phase III clinical trials, the efficacy against COVID-19 is almost more than 50%, and the protective effect against symptomatic and asymptomatic infections is high [1]. However, with the popularization of vaccines, people are aware of many potential problems. The production of vaccine-specific antibodies and the durability of acquired immunity are often not satisfactory, such that booster vaccines are required. Therefore, many people refuse to be vaccinated, bringing new public health problems.

Vaccines provide protection by generating specific antibodies. Accumulating evidence shows that T follicular helper (Tfh) cells play a fundamental role in the humoral immune response after vaccination and support the production of immunoglobulins (Ig) by B cells [2,3]. Therefore, it is not difficult to understand the role of the Tfh cells in the pathogenesis of SARS-CoV-2 infection and the COVID-19 vaccines [4,5,6,7]. However, how to regulate the function of Tfh cells to improve the effectiveness of the SARS-CoV-2 vaccines and the persistence of immunoglobulin production specific to SARS-CoV-2 has yet to be studied. In this review, we summarize the role of Tfh cells in mediating antibody production and the recent findings on the relationship between Tfh cells and the effectiveness of vaccines. We also discuss how to improve the effectiveness of the SARS-CoV-2 vaccines by regulating Tfh cells.

## 2. Tfh Cells Supporting Immunoglobulin Production

Tfh cells were first identified in human tonsils [8]. These cells support the activation of B cells, generation of plasma cells, and antibody responses in the germinal center (GC). B-cell lymphoma-6 (Bcl-6) is the main transcription factor of Tfh cells and promotes the expression of C-X-C chemokine receptor type 5 (CXCR5) [9,10,11]. CXCR5 is specifically expressed on the surface of Tfh cells and works as an attractant for B cell follicles to make intercellular contacts with B cells. Tfh cells produce interleukin (IL)-21 to promote the maturation of B cells and the production of antibodies [8].

Tfh cells are known for their high flexibility and plasticity [12]. While expressing Bcl-6, these cells can simultaneously express the transcription factors of other T helper (Th) cell subgroups, such as T-box factor (T-bet) for Th1 cells, GATA-binding protein 3 (GATA3) for Th2 cells, or retinoic-acid-receptor-related orphan nuclear receptor gamma (RORγT) for Th17 cells. We define these biphenotypic cells as different Tfh cell subsets, called Tfh1 cells, Tfh2 cells, and Tfh17 cells, respectively [12]. Each Tfh cell subset follows different developmental pathways due to environmental stimuli, resulting in functional heterogeneity. The functional diversity of Tfh cell subsets supports the isotype switching of GC B cells to induce different types of immunoglobulins [12].

## 3. Different Immunoglobulins Are Promoted by Tfh Cell Subsets

The immunoglobulin family contains five classes (IgM, IgA, IgE, IgG, and IgD) with different properties and functions [13]. IgM is the first immunoglobulin expressed during B cell development and is associated with primary immune responses. IgA is essential for protecting mucosal surfaces from toxins, viruses, and bacteria. IgE participates in hypersensitivity, allergic reactions, and reactions to parasitic infections. IgG is functionally divided into four subclasses, IgG_1_, IgG_2_, IgG_3_, and IgG_4_, and plays a crucial role in antiviral and vaccine responses. IgD is found at very low levels with a short serum half-life; its function is unclear [13] (Figure 1).

### 3.1. Tfh1 Cells Induce IgG_1_ and IgG_2_ Production after Infection and Vaccination

IgG_1_ is the most abundant immunoglobulin subclass, representing 60–70% of the total IgG [14]. IgG_1_ deficiency accounts for about 30% of all immunoglobulin deficiency diseases, leading to the aggravation of infection [15]. IgG_1_ mainly reacts to protein antigens, especially from viruses and vaccines [16,17]. During viral infection or after vaccination, interferons are secreted in large quantities, and the type I immune response is activated, resulting in the expansion of Tfh1 cells. Increased Tfh1 cells promote GC B cells to produce IgG_1_ [18,19]. In addition, pathological increases in IL-21 secretion and Tfh1 cell differentiation have been observed in some autoimmune diseases, such as systemic lupus erythematosus [20].

The IgG_2_ subclass accounts for 20–30% of total IgG, with an average concentration of 3 mg/mL in adult serum. IgG_2_ reacts to encapsulated bacteria and plays a key role in the humoral response towards polysaccharide antigens [14]. In interferon-deficient mice, the differentiation of Tfh1 cells is impaired, leading to a decrease in GC B cells and IgG_2a_ production [21,22,23].

### 3.2. Tfh2 Cells Induce IgG_4_ and IgE in Allergic Responses

The IgG_4_ subclass accounts for only 5% of the total IgG [14]. IgG_4_-related disease is a newly discovered allergic disease characterized by increased serum IgG_4_ levels and the infiltration of IgG_4_^+^ plasma cells into tissues [24]. A study showed that, compared to the control group, the frequency of Tfh2 cells was significantly increased in patients with IgG_4_-related disease and positively correlated with the serum levels of IgG_4_ and IL-4 [25].

IgE binds to the high-affinity receptor FcεRI2 expressed on the surface of mast cells and basophils, and crosslinks allergens, resulting in cell activation and the release of allergy mediators [26]. Tfh2 cells can promote B cell class conversion and induce IgE production [27]. In a mouse model of allergic asthma, allergens could mediate a type II immune response and the activation of Tfh2 cells [28,29]. Tfh cell deficiency can lead to a significant decrease in IgE production [30].

### 3.3. Tfh2 and Tfh17 Cells Promote IgA Secretion in Mucosal Immunity

The level of IgA in serum is lower than IgG, while secretory IgA is much higher on the surface of mucous membranes [4]. Secretory IgA can prevent pathogens from invading the mucosa [31]. Intestinal Peyer patches (PPs) are important lymphoid organs that produce secretory IgA to support the stabilization of the intestinal mucosal immunity [32]. Tfh17 cells have the potential to convert to Tfh cells in the PPs and support the production of IgA in the gut [33]. Abnormal deposition of serum glycosylated IgA_1_ on the walls of small vessels leads to IgA vasculitis (IgAV) [34,35]. A recent study showed that Tfh2 and Tfh17 cells increased in patients with IgAV compared to the control group and decreased during disease remission. Moreover, the frequencies of circulating Tfh2 cells and Tfh17 cells were positively correlated with serum IgA levels in IgAV [36]. However, the microenvironment conditions conducive to Tfh17 differentiation are still unclear.

## 4. Tfh Cell Response in SARS-CoV-2 Infection

Tfh cells protect against infectious diseases by promoting antibody responses to viral, bacterial, parasitic, and fungal infections [8]. Although it is possible to produce antibodies against SARS-CoV-2 in the absence of Tfh cells, optimal protection in vivo still relies on Tfh cells to produce high-affinity antibodies [37]. Studies suggest that Tfh cells promote GC responses and drive B cell differentiation to produce high-quality neutralizing antibodies, which is necessary to control SARS-CoV-2 infection [38]. Increased numbers of spike (S)-specific antibodies, memory B cells, and activated circulating Tfh (cTfh) cells, along with IgM and IgG antibodies bound to SARS-CoV-2 were detected in the blood of patients with mild and convalescent COVID-19 and the lymphonodus of rhesus monkeys infected with SARS-CoV-2 [39,40,41]. Lipsitch et al. found that cross-reactive memory Tfh cells can trigger faster and better antibody responses, accelerating the control of upper respiratory tract and lung viruses [42]. Moreover, IgG antibodies, neutralizing plasma cells, memory B cells, and memory T cells specific to SARS-CoV-2 are continuously produced by recovered patients for at least three months after infection and the receptor-binding domain-specific memory B cells and S-specific memory B cells increase over time [43,44]. Furthermore, virally infected Th1 polarization conditions usually increase cTfh1 cells in humans and rhesus monkeys infected with SARS-CoV-2 [40,41,45,46,47].

A general deficiency of T and B cells in lung tissues and dysregulation of the humoral immune response have been found in patients who died from COVID-19. Compared to focal tissue pneumonia, the formation of Tfh cells and reproductive centers were largely absent in the draining lymph nodes of late COVID-19 patients [48]. Kaneko et al. analyzed the thoracic lymph nodes and spleens of patients who died from acute SARS-CoV-2 infections and found that although these patients had a strong T-cell-mediated B cell response, the GCs were significantly deficient [49]. Long-lasting GCs can produce a long-term memory immune response and produce high-quality antibodies after exposure to antigens [50]. In the absence of GCs, Tfh cells are not generated to activate GC B cells. Non-GC B cell responses cannot induce long-term memory or the production of high-quality antibodies. Therefore, Tfh cells are required for the GC B cell response to SARS-CoV-2 infection [49].

## 5. Tfh Cell Responses in SARS-CoV-2 Vaccination

Due to the prominent function of Tfh cells in promoting B cell maturation and immunoglobulin production, these cells may play an important role in vaccine-mediated acquired immunity. Researchers evaluated the phenotypes of T cells from individuals who received the BNT162b2 mRNA vaccine and found that the S-specific Tfh cells in the blood peaked one week after the second immunization but persisted at nearly constant frequencies in the axillary lymph nodes for at least six months [51]. The frequencies of the Tfh cells in lymph nodes are associated with antigen-specific GC B Cells [52]. Moreover, immunocompromised individuals (e.g., kidney transplant recipients) fail to produce Tfh cells, demonstrating a defect in GC B cell responses specific to SARS-CoV-2 after vaccination [52]. Cavazzoni et al. disrupted the generation of Tfh cells in mice vaccinated with the SARS-CoV-2 spike protein vaccine and found that the frequency of GC B cells decreased by about 50%, indicating that Tfh cells are critical for a strong GC response [53].

In contrast, the expansion of Tfh cells may help improve the strength of the GC response and vaccine effectiveness. Alameh et al. suggested that lipid nanoparticle (LNP) preparations with intrinsic adjuvant activity could induce Tfh cells, GC B cells, long-lived plasma cells, and memory B cells, and increase the production of persistent, protective antibodies [54]. In addition, antigen re-exposure can establish and recall SARS-CoV-2 spike epitope-specific CD4^+^ T cell memory and induce the accumulation of Tfh cells, which may contribute to long-term protection against SARS-CoV-2 [55].

## 6. Factors Affecting Vaccine Effectiveness by Regulating Tfh Cell Responses

Because sufficient induction of the Tfh cell response is essential for the effectiveness of vaccines, it is necessary to identify the factors that affect the Tfh cell response after vaccination. In addition to the type of vaccine and adjuvant, the age, body microenvironment, and basic immune status of the vaccinees may also affect the Tfh cell response and alter the vaccine effect (Figure 2).

### 6.1. Age-Dependent Immune Response

Although the Tfh B cell responses in vaccinees can be generally activated, the degree of activation varies among different populations. Studies have shown that the effectiveness of a vaccine is related to the age of the vaccinees. In a prospective longitudinal cohort study, antibody responses after the first vaccination with the BNT162b2 mRNA vaccine were inversely proportional to age, and the antibody response was significantly higher in younger people than in older people [56]. A second dose of the vaccine can trigger high concentrations of IgG antibodies in the elderly population; however, these antibodies have weaker neutralization capacities [56].

Live-attenuated influenza vaccine can induce Tfh cell activation and long-term antibody responses in children; however, these effects are less in adults [57]. In a study of the 2014–2015 trivalent seasonal influenza vaccination in different age groups, there was a five-fold reduction in vaccine-specific antibodies in the serum of older adults compared to younger adults. The study showed that the deficiency in antibody production was associated with decreasing cTfh cells in the aged [58]. In aged mouse experiments, a reduction in Tfh cell responses was associated with the impaired T cell initiation induced by conventional type 2 dendritic cells (cDC2s). Treatment with Toll-like receptor 7 agonists can increase the number of antigen-carrying cDC2s and boost Tfh cell response in aged mice [58]. Therefore, we suggest that age may affect the Tfh cell response and vaccine effects, and age-related defects can be reversed by agonists.

### 6.2. Polarity of Tfh Cells Affects the Vaccine Effect

The dominant cytokine milieu at the earliest stage of differentiation determines the heterogeneity of the phenotype and function of Tfh cells. Thus, the body microenvironment may affect the role of terminal Tfh cells with different phenotypes in the immune response [12]. The Tfh1 subset causes isotype switching of GC B cells to produce IgG_1_, the major immunoglobulin for antiviral effects [12]. Therefore, a body microenvironment that promotes the differentiation of Tfh1 cells but not Tfh2 or Tfh17 cells would be conducive to high vaccine effectiveness.

After SARS-CoV-2 vaccination, there is an increase in the numbers of Tfh and Tfh1 cells in the peripheral blood of vaccinees [59]. Lederer et al. reported that SARS-CoV-2 mRNA vaccines could induce a stronger immune response than the rRBD-AddaVax vaccine. Moreover, they observed that bona fide Tfh cells induced by SARS-CoV-2 mRNA vaccines had a mixed Th1-Th2 functional profile, while rRBD-AddaVax triggered a Th2-skewed Tfh cell profile [60].

Nevertheless, a study of the low response to hepatitis B vaccination showed that although the number of total circulating Tfh cells was significantly lower in low responders, it was not due to an insufficient Tfh1 subpopulation. In contrast, cTfh2 and cTfh17 cells were severely shifted toward a cTfh1 phenotype in low responders [61]. Moreover, they demonstrated that miR-19b-1 and miR-92a-1 were associated with cTfh cell subset distribution and antibody production [61]. Above all, the shifts in Tfh cell subsets were closely associated with protective antibody response production. Thus, an increase in total Tfh cells alone may not be sufficient to improve antibody responses.

### 6.3. Initial Immune Status of Vaccines and Tfh Cell Responses

Many studies have confirmed that immunocompromised patients have a very high risk of developing COVID-19 disease and dying after contracting SARS-CoV-2 [62,63,64,65]. It is critical to protect these patients against COVID-19 by effective SARS-CoV-2 vaccination. Hagin et al. assessed the immunogenicity of the BNT162b2 vaccine in twenty-six patients with congenital immunodeficiency (IEI), eighteen of whom could produce a vaccine-specific antibody response after two doses of the vaccine [66]. Four of the patients were unable to achieve seroconversion. Delmonte et al. conducted a large study in which 85% of IEI patients had detectable antibodies after immunization; however, their antibody titers were significantly lower than healthy controls [67]. In acquired immunodeficiency, patients with advanced HIV infection also have a lower response to mRNA vaccination [68,69].

In most patients with autoimmune or autoinflammatory disease, the mRNA BNTb262 vaccine is immunogenic with a reliable safety profile. In patients treated with anti-CD20 or mycophenolate mofetil, BNT162b2-induced immunogenicity is significantly suppressed [70]. Moreover, it has been reported that, compared to control groups, the Tfh cell response after SARS-CoV-2 mRNA vaccination was lower in patients with multiple sclerosis who received anti-CD20 therapy [71]. Consequently, immune deficiency and treatment with immunosuppressants can attenuate Tfh cell responses, thereby reducing the effectiveness of vaccines.

### 6.4. Repeated Immunization Enhances the Intensity of the Tfh Cell Response

Repeated immunization with booster vaccines can improve the responsiveness of vaccinees by enhancing the Tfh cell response [55]. In a study of healthy subjects both naïve to and recovered from SARS-CoV-2 (recovered subjects monitored before and after mRNA prime and boost vaccination), all subjects naïve to SARS-CoV-2 showed robust T cell responses after the first dose of the vaccine that was further enhanced by the second dose [72]. In comparison, subjects who had recovered from SARS-CoV-2 had maximal T cell responses after the first vaccine dose, with little additional T cell boosting after the second dose [72].

Studies have shown that a subset of patients who failed to achieve seroconversion or had a low response after two mRNA vaccine doses responded to a third vaccine dose [73,74]. Kamar et al. [75] studied a cohort of thirty-seven solid organ transplant recipients, including thirty-one patients who had no response to the first three vaccine doses and five patients who responded weakly. Thirteen patients who did not respond to the first three doses were seroconverted with the fourth dose. Moreover, the fourth vaccine dose increased median antibody concentrations from 4 BAU/mL to 402 BAU/mL for the five patients who responded weakly to the first three vaccine doses [75].

### 6.5. Types of Vaccines and Inoculated Species

Studies have shown that the effects of vaccines vary according to the inoculated species. The NVX-CoV2373 vaccine, a SARS-CoV-2 subunit vaccine from the full-length spike protein, can elicit multifunctional CD4^+^ and CD8^+^ T cells, Tfh cells, and antigen-specific GC B cells in the spleens of mice [76]. SARS-CoV-2 spike and receptor-binding domain (RBD) immunogens produce robust B and Tfh cell responses in macaques. However, the spike protein subunit vaccine, not the RBD protein subunit vaccine, is potently immunogenic in mice [77]. In contrast, a protein subunit vaccine composed of the spike ectodomain protein (StriFK) plus a nitrogen bisphosphonate-modified zinc-aluminum hybrid adjuvant (FH002C) can generate substantially higher neutralizing antibody titers in mice, hamsters, and cynomolgus monkeys [78]. Compared to a recombinant protein vaccine formulated with the MF59-like adjuvant, a single immunization with SARS-CoV-2 mRNA elicited potent Tfh cell and GC B responses with superior activity specific to SARS-CoV-2 in mice [59]. The mRNA-1273 vaccine provided rapid protection in the upper and lower airways from SARS-CoV-2 infection in rhesus macaques by inducing Th1 and Tfh cell responses and eliciting robust SARS-CoV-2 neutralizing activity [79].

The results from clinical trials indicated that multiple inactivated SARS-CoV-2 vaccines, including BBIBP-CorV, CoronaVac, and BBV152, were safe, well-tolerated, and could induce satisfactory high neutralizing antibody titers to reduce the number of severe cases [80,81,82,83]. The BNT162b2 SARS-CoV-2 mRNA vaccine significantly increased cTfh cells, and the frequency of cTfh cells was positively correlated with anti-S-specific IgA and IgG antibody titers [84]. Tfh cell responses according to the specific vaccines are shown in Table 1.

## 7. Discussion

The persistence of the humoral response induced by the SARS-CoV-2 natural infection is relatively limited, and herd immunity to COVID-19 may be difficult to achieve through natural infection [85]. Therefore, we urgently need to control the epidemic through extensive SARS-CoV-2 vaccination. The effectiveness of the vaccine not only directly affects the anti-infection ability of the vaccinators but also indirectly affects the vaccination willingness and the vaccination rate of the population. We believe that Tfh cell activation levels can reflect vaccine effectiveness. Detecting the Tfh cell response after vaccination can predict the production of long-term high-affinity antibodies. Any factor that interferes with this response may potentially change the effectiveness of the vaccine. Factors that regulate the Tfh cell response may be potential targets for improving vaccine efficacy.

## Figures and Tables

**Figure 1 vaccines-10-01623-f001:**
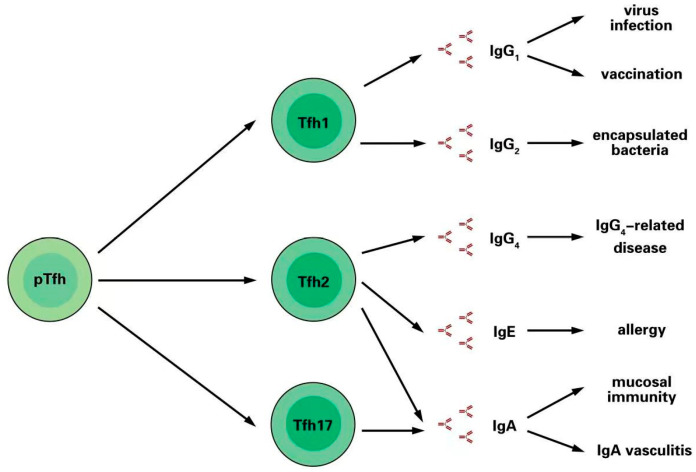
Tfh cell subsets support the production of immunoglobulins with different functions. Pre-Tfh (pTfh) cells can differentiate into Tfh1, Tfh2, or Tfh17 phenotypes under different microenvironmental conditions due to different physiological or pathological states. Each Tfh cell subset is functionally specific and can mediate B cell differentiation to produce different types of immunoglobulins that participate in various immune responses.

**Figure 2 vaccines-10-01623-f002:**
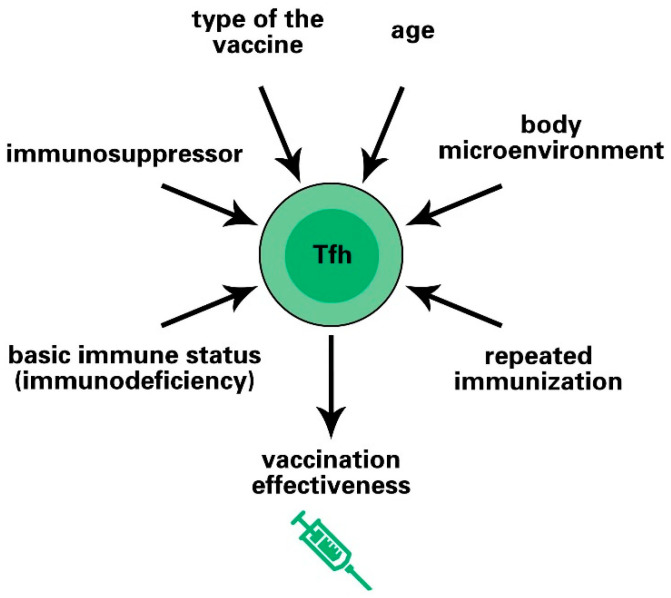
Simplified illustration of the factors that affect vaccine effectiveness by modulating Tfh cell response. Vaccine effectiveness is dependent on the activation levels of Tfh cells. Tfh cell responses can be altered by various factors, including the type of vaccine, age, body microenvironment, repeated immunization, basic immune status, and immunosuppressors.

**Table 1 vaccines-10-01623-t001:** Tfh cell responses to various types of SARS-CoV-2 vaccines.

	Vaccine Types	Species	Location	Tfh-Related Response	Antibody	Reference
1	NVX-CoV2373 protein vaccine	mice, baboons	blood, spleen	Elicits multifunctional CD4^+^ and CD8^+^ T cells, Tfh cells, and antigen-specific GC B cells	IgG	[75]
2	Spike protein subunit vaccine and Receptor-Binding Domain protein subunit vaccine	macaques	blood, lymph node	Elicits potent antibody responses and serum neutralizing activity	IgG	[76]
3	Spike protein subunit vaccine	mice	blood, lymph node	Elicits robust GC B cell and Tfh cell responses	IgG	[76]
4	StriFK-FH002C protein vaccine	mice, hamsters, cynomolgus monkeys	blood, lymph node	Generates substantially higher neutralizing antibody titers	IgG, IgG1, IgG2a, IgG2b	[77]
5	Protein vaccine formulated with MF59-like adjuvant	mice	blood	Triggers Th2-skewed Tfh cell profile	IgG1	[59]
6	BBIBP-CorV protein vaccine	human	blood	Induces rapid, robust humoral responses	No data	[79]
7	CoronaVac protein vaccine	human	blood	Induces rapid, robust humoral responses	IgG	[80]
8	BBV152 protein vaccine	human	blood	Induces high neutralizing antibody responses; enhances humoral and cell-mediated immune responses	IgG	[82]
9	SARS-CoV-2 mRNA vaccine	mice	blood, lymph node, bone marrow, spleen	Elicits potent GC B and Tfh cell responses with superior ability specific to SARS-CoV-2	IgG1, IgG2a, IgG2b	[59]
10	mRNA-1273 vaccine	rhesus macaques	blood, lung	Induces Th1 and Tfh cell responses; elicits robust SARS-CoV-2 neutralizing activity	IgA, IgG	[78]
11	BNT162b2 mRNA vaccine	human	blood	Induces cTfh cells; the frequency of cTfh cells is positively correlated with anti-Spike-specific IgA and IgG antibody titers	IgA, IgG	[83]

## Data Availability

Not applicable.

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
