# Peer review of "Characteristics and Roles of T Follicular Helper Cells in SARS-CoV-2 Vaccine Response"

_vaccines, 2022, doi:10.3390/vaccines10101623_

Round 1

Reviewer 1 Report

The manuscript by Chi et al., tries to summarize the role of Tfh cells in SARS-CoV-2 infection and vaccination. However, several key publications are missing throughout the manuscript, which therefore requires substantial revision. In addition, the manuscript would clearly benefit from improving the use of the English language.

Ref. 3 is only one of the two original papers that first described Tfh cells. Either also cite PMID: 11104798 or only cite recent reviews (in addition to ref 2). Actually, beside ref 2, there have already been several reviews published before by other groups that covered the topic of the manuscript under review here and they should be considered as well: PMID: 33788271, PMID: 34603308, PMID: 35909969.

Line 54: Besides Ref 5 and 6, PMID: 19608860 needs also to be cited as one of the three hallmark papers that first described functional Bcl6 requirements for Tfh cells.

Refs 4 and 7 are from the same author and redundant. It would be sufficient to cite the newer review (ref 7).

Lines 121-132: A recent paper (PMID: 34914544) showed that high-affinity antibody responses against SARS-CoV-2 can also occur in the absence of Tfh cells. This study needs to be discussed here as well.

The first study to describe circulating Tfh cells in more detail in COVID-19 is missing here: “Humoral and circulating follicular helper T cell responses in recovered patients with COVID-19” (PMID: 32661393).

Lines 154-173: Several key papers on the role of Tfh cells in SARS-CoV-2 infection and vaccination are missing and need to be discussed:

“SARS-CoV-2 mRNA vaccination elicits a robust and persistent T follicular helper cell response in humans” (PMID: 35026152). Elegant study by the Ellebedy lab using fine-needle aspiration of draining lymph nodes from individuals who received the BNT162b2 mRNA vaccine.

Germinal center responses to SARS-CoV-2 mRNA vaccines in healthy and immunocompromised individuals (PMID: 35202565)

Lipid nanoparticles enhance the efficacy of mRNA and protein subunit vaccines by inducing robust T follicular helper cell and humoral responses (PMID: 34852217)

Establishment and recall of SARS-CoV-2 spike epitope-specific CD4+ T cell memory (PMID: 35314848

Line 189: “the antibody response after the first vaccination with the BNT162b2 COVID-19 vaccine was proportional to the age,…” That is not correct, it was “inversely proportional”

Line 210: A reference(s) is required for this statement: “Tfh1 subset leads to isotype switching of GC B cells to produce IgG1, which is the major immunoglobulins for antiviral effects.”

Line 243: This is misleading: “In patients treated with 243 corticosteroids (anti-CD20 or mycophenolate mofetil),…” anti-CD20 nor MM are corticosteroids. Clarify.

Author Response

The manuscript by Chi et al., tries to summarize the role of Tfh cells in SARS-CoV-2 infection and vaccination. However, several key publications are missing throughout the manuscript, which therefore requires substantial revision. In addition, the manuscript would clearly benefit from improving the use of the English language.

Response: We deeply appreciate the reviewer for careful evaluation and valuable his/her comments. Due to the time limit of submission, we asked native English speaker to do a proofreading to improve the manuscript text. The following text describes our response to the comments made by the reviewer.

Ref. 3 is only one of the two original papers that first described Tfh cells. Either also cite PMID: 11104798 or only cite recent reviews (in addition to ref 2). Actually, beside ref 2, there have already been several reviews published before by other groups that covered the topic of the manuscript under review here and they should be considered as well: PMID: 33788271, PMID: 34603308, PMID: 35909969.

Response: We deeply appreciate the reviewer’s valuable comment. According to the comment, we added PMID: 11104798, PMID: 33788271, PMID: 34603308, and PMID: 35909969. We revised the manuscript in section 1 as follow “Accumulating evidences show that T follicular helper (Tfh) cells play a fundamental role in humoral immune response after vaccinations and support the production of immunoglobulin (Ig) by B cell [ J Exp Med 2000,192,1545-52; 53-62]. Therefore, it is not difficult to understand the role Tfh cells in the pathogenesis of SARS-CoV-2 infection and in the COVID-19 vaccine [Curr Opin Immunol 2022,74,112-7; Eur J Immunol 2021,51,1325-33; Front Immunol 2021,12,731100; Front Cell Infect Microbiol 2022,12,953022.].”. (Revised version page 1, line 42)

Line 54: Besides Ref 5 and 6, PMID: 19608860 needs also to be cited as one of the three hallmark papers that first described functional Bcl6 requirements for Tfh cells.

Response: We thank the reviewer for very insightful comments. As advised, we added PMID: 19608860. We revised the manuscript in section 2 as follow “The main transcription factor of Tfh cells is B cell lymphoma-6 (Bcl-6), which promote the expression of C-X-C chemokine receptor type 5 (CXCR5) [ Science 2009,325,1001-5; 6-10; Immunity 2009,31,457-68.].”. (Revised version page 2, line 55)

Refs 4 and 7 are from the same author and redundant. It would be sufficient to cite the newer review (ref 7).

Response: We agree with the reviewer for suggestion. According to the comment, we removed ref 4 and cited ref 7 (ref 8 in revision). (Revised version page 2, line 55 and 59)

Lines 121-132: A recent paper (PMID: 34914544) showed that high-affinity antibody responses against SARS-CoV-2 can also occur in the absence of Tfh cells. This study needs to be discussed here as well.

Response: We agree with the reviewer’s opinion. As advised, we added PMID: 34914544, and revised the manuscript in section 4 as follow “Although it is also possible to produce non-Tfh-dependent high affinity antibodies against SARS-CoV-2 in the absence of Tfh cells, optimal protection in vivo still relies on Tfh cells to produce high affinity antibodies [Sci Immunol 2022,7,eabl5652.].”. (Revised version page 3, line 122)

The first study to describe circulating Tfh cells in more detail in COVID-19 is missing here: “Humoral and circulating follicular helper T cell responses in recovered patients with COVID-19” (PMID: 32661393).

Response: We agree with the reviewer’s opinion. As advised, we added PMID: 32661393, and revised the manuscript in section 4 as follow “Increased numbers of Spike (S)-specific antibodies, memory B cells, and activated circulating Tfh (cTfh), as well as SARS-CoV-2-bound IgM and IgG antibodies, were detected in the blood of patients with mild and convalescent COVID-19 and in the lymphonodus of rhesus monkeys infected with SARS-CoV-2 [Nat Med 2020,26,453-5; 1428-34; Nat Commun 2021,12,541.].”. (Revised version page 3, line 127)

Lines 154-173: Several key papers on the role of Tfh cells in SARS-CoV-2 infection and vaccination are missing and need to be discussed: “SARS-CoV-2 mRNA vaccination elicits a robust and persistent T follicular helper cell response in humans” (PMID: 35026152). Elegant study by the Ellebedy lab using fine-needle aspiration of draining lymph nodes from individuals who received the BNT162b2 mRNA vaccine.

Response: We agree with the reviewer’s opinion. As advised, we added PMID: 35026152, and revised the manuscript in section 5 as follow “Researchers evaluated the phenotypes of T cells from individuals who received BNT162b2 mRNA vaccine, and found that the S-specific Tfh cells in the blood peaked one week after the second immunity, while persisted at nearly constant frequencies in axillary lymph nodes for at least 6 months [Cell 2022,185,603-13.e15.].. (Revised version page 4, line 152)

Germinal center responses to SARS-CoV-2 mRNA vaccines in healthy and immunocompromised individuals (PMID: 35202565)

Response: We thank the reviewer for the comment. As suggested, we added PMID: 35202565, and revised the manuscript in section 5 as follow “Further study showed that frequencies of Tfh cells in lymph nodes were associated with antigen-specific GC B Cells. Moreover, different from healthy individuals, immunocompromised individuals such as kidney transplant recipients failed to produce Tfh cells, and therefore showed a defect in SARS-CoV2-specific GC B cell responses after vaccination [Cell 2022,185,1008-24.e15.].”. (Revised version page 4, line 155)

Lipid nanoparticles enhance the efficacy of mRNA and protein subunit vaccines by inducing robust T follicular helper cell and humoral responses (PMID: 34852217)

Response: We agree with the reviewer’s opinion. As advised, we added PMID: 34852217, and revised the manuscript in section 5 as follow “Alameh et al. suggested that lipid nanoparticle (LNP) preparations with intrinsic adjuvant activity could promote the induction of Tfh cells, GC B cells, long-lived plasma cells and memory B cells, and increase the production of persistent and protective antibodies [Immunity 2022,55,1136-8.].. (Revised version page 4, line 164)

Establishment and recall of SARS-CoV-2 spike epitope-specific CD4+ T cell memory (PMID: 35314848)

Response: We agree with the reviewer for suggestion. According to the comment, we added PMID: 35314848, and revised the manuscript in section 5 as follow “In addition, antigen re-exposure can establish and recall SARS-CoV-2 spike epitope-specific CD4+T cell memory and induce the accumulation of Tfh cells, which may contribute to long-term protection against SARS-CoV-2 [Nat Immunol 2022,23,768-80.].”. (Revised version page 4, line 167)

Line 189: “the antibody response after the first vaccination with the BNT162b2 COVID-19 vaccine was proportional to the age,…” That is not correct, it was “inversely proportional”

Response: We apologize for the error. We have revised the manuscript in section 6.1 as follow “In a prospective longitudinal cohort study, the antibody response after the first vaccination with the BNT162b2 COVID-19 vaccine was inversely proportional to the age, and the antibody response was significantly higher in young people than in older people.”. (Revised version page 4, line 180)

Line 210: A reference(s) is required for this statement: “Tfh1 subset leads to isotype switching of GC B cells to produce IgG1, which is the major immunoglobulins for antiviral effects.”

Response: We deeply appreciate the reviewer’s comment. According to the comment, we added PMID: 33717120 (Front Immunol 2021,12,621105.) as the reference. (Revised version page 5, line 201)

Line 243: This is misleading: “In patients treated with 243 corticosteroids (anti-CD20 or mycophenolate mofetil),…” anti-CD20 nor MM are corticosteroids. Clarify.

Response: We apologize for the error. We have revised the manuscript in section 6.4 as follow “In patients treated with anti-CD20 or mycophenolate mofetil, BNT162b2-induced immunogenicity is significantly reduced.. (Revised version page 5, line 232)

Reviewer 2 Report

After reviewing the paper submitted for publication as a review, I consider the review of the role of follicular T helper cells in the immune response to SARS-CoV-2 vaccination to be adequate and sufficiently developed. The clinical readership would undoubtedly like to know more about the clinical implications for vaccination and the development of new therapeutic targets, which, although the authors comment in the text, I believe should be extended a little more. With this modification, perhaps of a clinical nature, I believe that the article could be published.

Author Response

After reviewing the paper submitted for publication as a review, I consider the review of the role of follicular T helper cells in the immune response to SARS-CoV-2 vaccination to be adequate and sufficiently developed. The clinical readership would undoubtedly like to know more about the clinical implications for vaccination and the development of new therapeutic targets, which, although the authors comment in the text, I believe should be extended a little more. With this modification, perhaps of a clinical nature, I believe that the article could be published.

Response: We were pleased to know that the reviewer agrees with us on the value of our study. We revised the manuscript for further improvement.

Reviewer 3 Report

The authors review the characteristics and roles of T follicular helper cells in SARS-2 CoV-2 vaccine response. Focusing on T cells after vaccination is worthy. However, some comments should be addressed to improve this manuscript.

1. There are many kinds of vaccines nowadays. Please summarize the Tfh cell response according to the specific vaccines widely used in the table for better readability.

2. Please add the methods for the response of Tfh cell after vaccination.

The adopted methods are important to assess the T cell response.

3. Regarding to the microenvironment, the detailed cytokines related to the Tfh cell should be described.

Author Response

The authors review the characteristics and roles of T follicular helper cells in SARS-2 CoV-2 vaccine response. Focusing on T cells after vaccination is worthy. However, some comments should be addressed to improve this manuscript.

Response: We deeply appreciate the reviewer for careful evaluation and valuable his/her comments. The following text describes our response to the comments made by the reviewer.

  1. There are many kinds of vaccines nowadays. Please summarize the Tfh cell response according to the specific vaccines widely used in the table for better readability.

Response: We thank the reviewer for the comment. According to the comment, we added Table, and added in section 6.5 in manuscript. (Revised version page 6, line 256; Table)

  1. Please add the methods for the response of Tfh cell after vaccination. The adopted methods are important to assess the T cell response.

Response: We appreciate the valuable comment. Based on the advice, we revised Table and added the sampling location for the response of Tfh cell after vaccination. (Revised version page 6, line 277; Table)

  1. Regarding to the microenvironment, the detailed cytokines related to the Tfh cell should be described.

Response: We deeply appreciate the reviewer’s comment. According to the comment, we revised the manuscript in section 3.1 as follow “It is reported that during virus infection or after vaccination, interferons were secreted in large quantities, and type I immune response was activated, resulting in the expansion of Tfh1 cells.” and “In addition, increased IL-12 secretion and pathological increasing of Tfh1 cells were also observed in some autoimmune diseases such as systemic lupus erythematosus.(Revised version page 2, line 84 and 87); Study showed that compared with control group, the frequency of Tfh2 cells significantly increased in patients with IgG4-related disease, and was positively correlated with the serum levels of IgG4 and IL-4.” and “However, the microenvironment conditions conducive to Tfh17 differentiation are still unclear.(Revised version page 3, line 98 and 118).

Round 2

Reviewer 1 Report

While I appreciate the addition of the missing references, the text still requires substantial English language editing.

Author Response

While I appreciate the addition of the missing references, the text still requires substantial English language editing.

Response: We sincerely thank reviewer for thoroughly examining our manuscript and providing very helpful comments to guide our revision. To improve the manuscript, we have looked for a professional editing company to make grammar revisions. Once again,thank you very much for your comments and suggestions.

Reviewer 3 Report

I found that the authors made effort to address the comments. I hope that this manuscript would contribute to understand T cells in SARS-CoV-2 vaccine response. 

Author Response

I found that the authors made effort to address the comments. I hope that this manuscript would contribute to understand T cells in SARS-CoV-2 vaccine response.

Response: We are extremely grateful of the reviewer for the constructive comments and suggestions. We appreciated for reviewer warm work earnestly.
